# Unique and universal dew-repellency of nanocones

Pierre Lecointre[1,2 ✉], Sophia Laney[3], Martyna Michalska [3], Tao Li[3], Alexandre Tanguy[4], Ioannis Papakonstantinou [3 ✉] & David Quéré [1,2 ✉]

Surface structuring provides a broad range of water-repellent materials known for their ability to reflect millimetre-sized raindrops. Dispelling water at the considerably reduced scale of fog or dew, however, constitutes a significant challenge, owing to the comparable size of droplets and structures. Nonetheless, a surface comprising nanocones was recently reported to exhibit strong anti-fogging behaviour, unlike pillars of the same size. To elucidate the origin of these differences, we systematically compare families of nanotexture that transition from pillars to sharp cones. Through environmental electron microscopy and modelling, we show that microdroplets condensing on sharp cones adopt a highly non-adhesive state, even at radii as low as 1.5 μm, contrasting with the behaviour on pillars where pinning results in impedance of droplet ejection. We establish the antifogging abilities to be universal over the range of our cone geometries, which speaks to the unique character of the nanocone geometry to repel dew. Truncated cones are finally shown to provide both pinning and a high degree of hydrophobicity, opposing characteristics that lead to a different, yet efficient, mechanism of dew ejection that relies on multiple coalescences.

[1] Physique et Mécanique des Milieux Hétérogènes, UMR 7636 du CNRS, ESPCI, PSL Research University, Paris, France. [2] LadHyX, UMR 7646 du CNRS, École Polytechnique, Institut Polytechnique de Paris, Palaiseau, France. [3] Photonic Innovations Lab, Department of Electronic and Electrical Engineering, University College London, London, UK. [4] Laboratoire de Mécanique des Solides, UMR 7649 du CNRS, École Polytechnique, Institut Polytechnique de Paris, Palaiseau, France. ✉email: pierre.lecointre@polytechnique.org; i.papakonstantinou@ucl.ac.uk; david.quere@espci.fr

Spontaneous jumping of condensing droplets[1] has recently emerged as a promising solution for antifogging applications[2–4], among many others[5–10]. For this to be achieved, droplets formed through condensation must exhibit large contact angles and minimal pinning to the substrate[11,12]. While this is considered a challenge for micrometre scale droplets, cicada wing-inspired surfaces with nanocone arrays[2,13,14] have been shown to exhibit dew-repellency and thus constitute a promising route to elicit special wetting properties at microscales.

During condensation, coalescence of neighbouring non-wetting droplets induces the conversion of surface energy into kinetic energy[1], which possibly promotes droplets to jump away from the surface, hence providing antifogging behaviour. The proportion $N$ of drops jumping after coalescence (rate of departure) is a measure of the antifogging efficiency, and it was found to exceed 90% on hydrophobic nanocones[14], instead of at best 35% on previously reported textured materials[14,15]. This spectacular property was assumed to originate from the combination of texture scale (sub-micrometre), shape (conical) and density (dense array), without however, systematic experiments to verify this hypothesis. Hence it appears crucial to investigate families of conical structures in order to establish the versatility and universality of the antifogging efficiency of nanocones, and additionally explore where the boundary in performance extends to. To that end, we build nanostructures from cylindrical to truncated and to conical (Fig. 1a) and consider specifically three families of nanocones: homothetic (differing in the pitch and height but with constant apex angle), extruded (differing in the height and apex angle but with constant pitch), and truncated (with a given design and different degrees of truncation).

We first evidence the unique microwetting properties of sharp nanocones after observing condensed droplets by environmental scanning electron microscopy (ESEM). Then, we focus on dew-repellency and quantitatively discuss its universality in a wide window of geometries. Truncated cones, however, behave differently and we show that their significant adhesion to microdrops does not prevent successful antifogging, owing to an efficient droplet ejection after triple, quadruple and fivefold coalescences.

## Results

### Imaging condensation at the microscale.
Using block-copolymer self-assembly and plasma etching, we design nine centimetre-size arrays of nanocones (height $h$) arranged on a dense hexagonal lattice (pitch $p$). We also employ two reference materials consisting of nanopillars (sample A) and nanocones (sample H1) to connect our findings to previous investigations[14,16]. Fig. 1 shows the sample library. Family H refers to homothetic texture where the index ranks the relative size of structures, from lowest to highest pitch $p$ (from 52 nm to 110 nm), at fixed aspect ratio $h/p = 2.2 \pm 0.2$. Family E is that of extruded cones, where materials are ranked from lowest to highest height (from 144 to 420 nm), at fixed $p = 110 \pm 5$ nm. The two families H and E intersect in one sample (H3/E2) with $p = 110$ nm and $h = 250$ nm. The cone sharpness $\Sigma = 1/2\tan^{-1}(p/2\,h)$, defined as the inverse of their apex angle $\beta$, varies in our study between 1 and 4. Finally, family T includes truncated cones with same pitch $p = 110 \pm 5$ nm as in E, and it is classified from smallest to largest top diameter $l$ (from 34 to 60 nm). Details about the samples and their fabrication are provided in the methods section. Next, the resulting surfaces are rendered hydrophobic by vapour deposition of 1H,1H,2H,2H-perfluorodecyltrichlorosilane. Such a treatment on flat silicon yields an advancing contact angle $\theta_a = 120° \pm 2°$, a value that greatly increases to $\theta_a = 166° \pm 5°$ upon nanostructuring.

The adhesion of water to its substrate is quantified by the contact-angle hysteresis, which we measure by slowly dispensing millimetre-size drops (Supplementary Table 1). On the one hand, hysteresis is ca. 10° on samples H and E, a small value compared to the contact angles—the hallmark of repellent materials. All samples H and E have sharp structures favouring a poor wetting, except E4 whose rounded and continuous top prevents contact lines from pinning, thereby providing wetting properties similar to sharper cones. On the other hand, hysteresis roughly triples to ca. 30° on nanopillars and truncated cones. We attribute this to the discontinuous edges at the top of these structures, which pins the contact line during receding motion. These differences can be amplified for microdrops: water condensing within nanopillars can remain trapped inside the vertical texture, which reinforces pinning and immobilises droplets[17]. In contrast, water was assumed to spontaneously leave the core of dense nanocones: in such an asymmetric landscape, the nucleus lowers its surface energy by rising-up the structure to sit atop the cones, in the so-called Cassie state[14,18–21]. The expulsion of water nuclei from the conical texture is especially difficult to monitor directly, due to both the size (of order $p$) of the nuclei and the short time (nanoseconds) anticipated for their displacement over the nanoscale height $h$. However, this scenario implies differences in the morphology of microdroplets growing on nanocones compared to those on nanopillars, and thus in their mobility—a property of paramount importance for antifogging.

Condensing microdroplets can be observed directly by ESEM, whose high resolution and image sharpness enables us to visualise drops in the early stage of condensation and to access contact angles on the microscale ($r > 350$ nm). The operating conditions are carefully optimised to minimise heating[22,23], contamination[24–26] and radiation damage[27] (Supplementary discussion and Supplementary Fig. 1). The sample holder (60°-tilted copper bracket) can accommodate a wide tilting range (up to 90°), crucial to render a clear view of the evolving/resting droplets. Furthermore, the bracket is mounted on a Peltier cooling stage and temperature and chamber pressure are controlled around $-2 \pm 1\,°C$ and $600 \pm 100\,Pa$, respectively.

Images of water condensing either on nanopillars (sample A, Fig. 2a) or nanocones (sample E4, Fig. 2b) reveal marked differences: for all drop sizes, the apparent contact angle of water is much larger on E4 than on A. Water even seems to "levitate" on nanocones, with corresponding angles of $171° \pm 4°$. Furthermore Fig. 2b shows a large collection of microdroplets (~70) all in this highly non-wetting state, and thus likely to be ultra-mobile despite their scale. These droplets are so close to being spherical that it proves extremely difficult to define a contact area. The micrographs captured on other nanocones, either sharp or truncated (Supplementary Figs. 2, 3, and 4), are similar to that in Fig. 2b, verifying this key observation applies to conical nanostructures.

The contact angle also increases with the drop size, regardless the sample type, which can be seen both by exploiting the droplet polydispersity (with radii spanning from ~1 μm to ~25 μm) displayed in Fig. 2a, b, or following individual condensation events, as pointed in Fig. 2c, d. In the first case, droplets with radius $r < 2$ μm have a typical contact angle of 120° on sample A and this value rises to ~140° for larger droplets. In the second case, the angle of a growing droplet also increases (Fig. 2c), with successive growth modes:[12,17] starting at $r \approx 0.85$ μm with $\theta = 120° \pm 5°$ (image 1), the nucleus retains a constant base area while its angle rises to $140° \pm 5°$ (images 2–3), a value maintained throughout (images 4–8). In contrast, the angle on surface E4 (Fig. 2d) rapidly increases from $130° \pm 7°$ (for $r \approx 0.6$ μm, image 2) to its final value of $171° \pm 4°$ ($t > 3.6$ s, $r > 1.2$ μm, images 3–6).

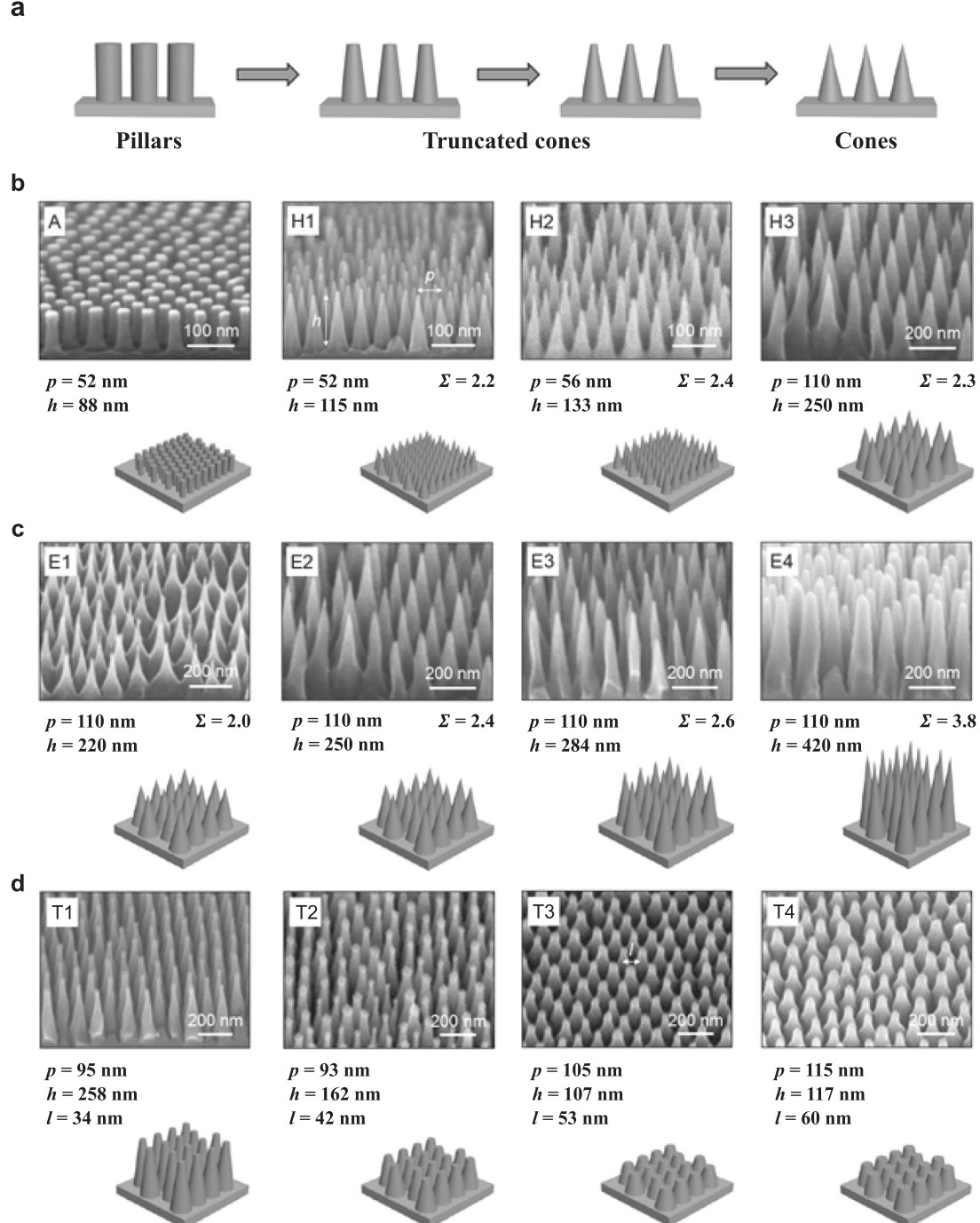

**Fig. 1 Families of samples. a** Schematic illustrating the geometry transition from nanopillar, to truncated cone and finally, to sharp cone. **b–d** Scanning electron microscopy (SEM) images and schematics corresponding to the three families of nanocones. For all surfaces, the cones with height $h$, pitch $p$, apex angle $\beta$ and sharpness $\Sigma = 1/\beta$ are arranged on a dense hexagonal array and coated by a hydrophobic layer. **b**. A is a reference sample made of nanopillars with diameter $l = 30$ nm. H1 is the smallest nanoconical texture, H2 and H3 are homothetic (constant $h/p$), with a size ratio of 1.1 and 2.1, respectively. **c** E1, E2, E3 and E4 are of equal pitch $p = 110$ nm and gradually extruded from E1 with $h = 144$ nm by a factor of 1.7, 2.0 and 2.9, respectively. Families H and E intersect: H3 and E2 are the same material. **d** T1, T2, T3 and T4 have the same pitch but are truncated, with various top diameters $l$.

Hence, we observe contrasting condensation patterns between both samples. On the one hand, contact angles on nanopillars are systematically smaller than those on nanocones and suffer from contact line pinning[12,17], two facts that express deep solid/liquid interactions. On the other hand, apart from a short transient state, droplets on nanocones rapidly exhibit very high, macroscopic-like, contact angles. This strongly suggests a Cassie state triggered at a radius of ~1 μm, a unique behaviour at the scale where water generally penetrates pillar-like structures.

This first series of experiments can be condensed into one graph, by plotting the contact angle as a function of the droplet radius $r$ (Fig. 3). Data are obtained by fitting the contour of drops by a circle of radius $r$ completed by a baseline with radius $\lambda = r\sin\theta$, so that these two independent measurements provide $\theta$. In

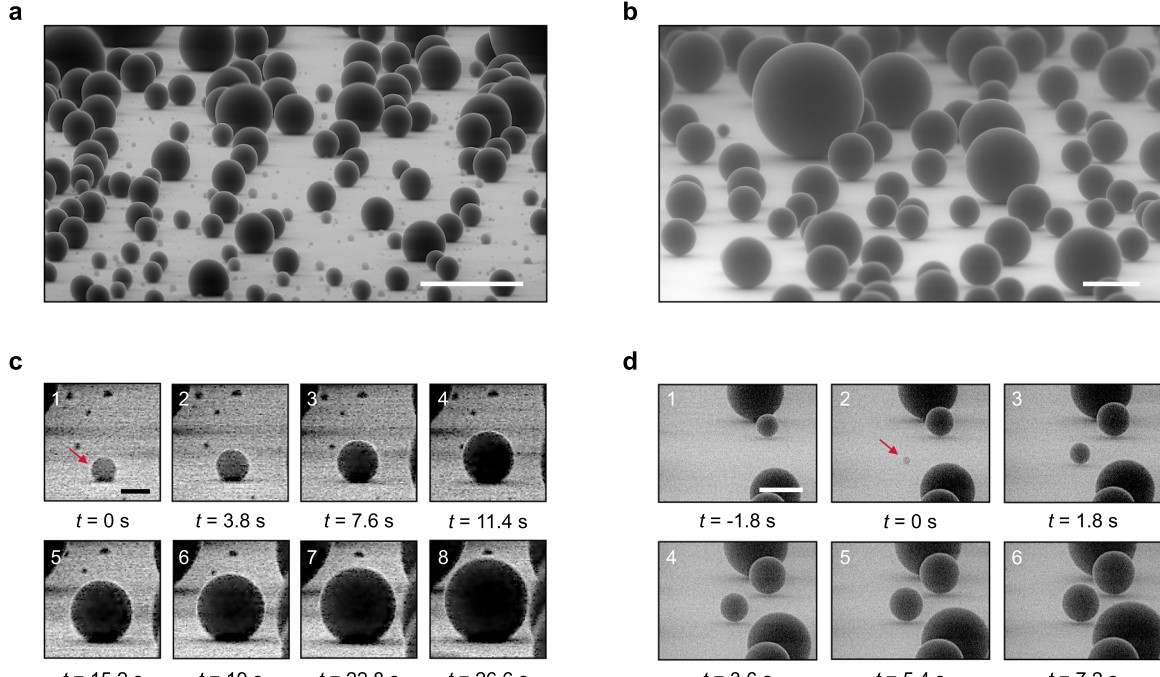

**Fig. 2 Direct visualisation of droplets condensing on nanotexture. a** ESEM images of water microdrops condensing on nanopillars (sample A; tilted by 85°). Drops adhere to the surface with contact angles no larger than 140°. The scale bar shows 20 μm. **b** ESEM images of microdrops condensing on nanocones (sample E4; tilted by 80°). Contact angles are now ~170° for all drops (radii $r$ between 1 μm and 23 μm). The scale bar shows 20 μm. **c** Growth dynamics of an individual droplet (pointed by the arrow) on sample A. The drop is first pinned (images 1–3) with a contact angle increasing from 120° to 140°, after which it keeps this value. Images are separated by 3.8 s, substrate temperature is $T_s = -2.5$ °C and chamber pressure $P = 600$ Pa. The scale bar shows 2 μm. **d** Growth dynamics of a nucleus on sample E4. Starting with a contact angle of 130° ± 7° (image 2), the droplet quickly becomes a quasi-sphere with an angle of 171° ± 4°, proving a Cassie state even at a microscale. Images are separated by 1.8 s, temperature is $T_s = -1.5$ °C and pressure $P = 700$ Pa. The scale shows 2 μm.

the experiments, the baseline progresses at a velocity $v = d\lambda/dt$ ranging between 0.3 μm/s and 1 μm/s. Hence the capillary number $\eta\varpi/\gamma$ (denoting $\eta$ and $\gamma$ as the viscosity and surface tension of water) is typically $10^{-8}$, indicating a quasi-static regime for the advancing angle $\theta_a$ of water microdrops.

We first comment the differences between pillars and cones. Fig. 3 consolidates the results noted in Fig. 2, in that there is a distinct difference in contact angles between the samples, amounting to ~30° smaller contact angles on nanopillars than on nanocones at all radii $r$. Contact angles on truncated cones are slightly smaller than on sharp cones: despite the presence of flat areas at the cone tops, they maintain the high values characteristic of a Cassie state. In addition, the effect of drop size is confirmed for all samples: as $r$ changes from micrometric to decamicrometric values, $\theta_a$ increases by ~30° and it plateaus at a value of $\theta_a = 141° \pm 3°$ on A, $\theta_a = 160° \pm 2°$ on T4 and $\theta_a = 171° \pm 3°$ on E4. Interestingly, these values differ from those measured with millimetric water drops, as shown by the dotted lines in Fig. 3. The discrepancy is especially large for sample A, where the "macroscopic" angle is $\theta_a = 167° \pm 3°$, a high-value typical of a Cassie state. This confirms our former hypothesis: unlike deposited millimetric drops, condensing droplets partially grow within the A-texture and thus coexist with trapped water, a situation that renders the substrate more hydrophilic. Yet, the substrate remains globally hydrophobic, suggesting that condensing drops are in a partial Cassie state (that is, coexisting with a mixture of trapped water and trapped air)[12,17,28,29]. At small radii, the lower contact angles agree with this scenario; if nuclei form inside the texture, the smaller the droplet, the more effectively hydrophilic the substrate.

At first glance, the situation with the nanocones is more surprising with a saturation value of the contact angle $\theta_a = 171° \pm 3°$ larger than the macroscopic angle $\theta_a = 164° \pm 3°$. The effect is modest, yet systematic (despite error bars), as if the material exhibited an augmented hydrophobicity for $r > 1.5$ μm, a property of obvious practical interest for anti-dew materials. At a millimetre-scale, gravity tends to flatten water, hence decreasing its apparent contact angle. The size of the gravity-driven contact scales as $r^2\kappa$ for a non-wetting drop, denoting $\kappa^{-1} = (\gamma/\rho g)^{1/2}$ as the capillary length, $\rho$ as the water density, and $g$ as the gravity acceleration[30]. Weight can be neglected provided we have $r \sin\theta > r^2\kappa$, that is, $r < \kappa^{-1} \sin\theta \approx 600$ μm. This condition is largely fulfilled in Figs. 2 and 3 for condensing drops, which can explain the difference between angles obtained at micro- and milli- scales. Macroscopic measurements of contact angles are performed with millimetric drops so that gravity increases the apparent solid/liquid contact, an artefact leading to an underestimation of high contact angles. This suggests that the genuine advancing angle is rather the one observed with condensing drops. For truncated cones, where angles are smaller, the discrepancy between micro- and milli-measurements is more modest, in good agreement with our arguments where the discrepancy increases with the value of the angle.

Small nuclei on nanocones also deserve a discussion. Below $r = 1.5$ μm (yet with $r > p$), the contact angle significantly decreases, which we interpret as an effect of Laplace pressure. To advance our understanding, we create a model for the depth of drop penetration within the structures, depending on the drop radius and on the cone geometry. The surface force opposing water penetration by distance $z$ scales as $\gamma\beta z$ per cone[31–33], where

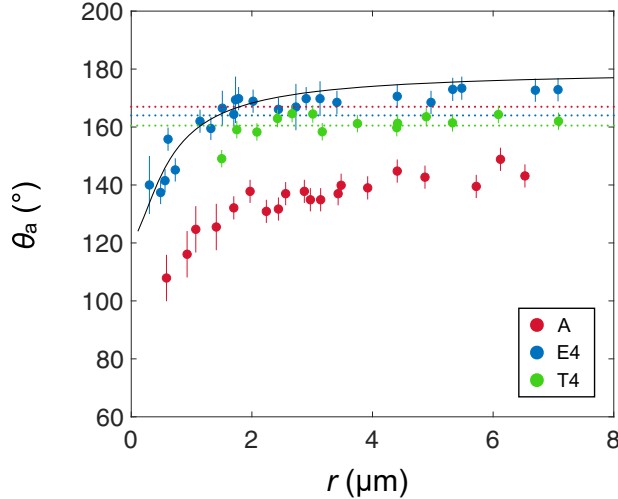

**Fig. 3 Contact angle of condensing microdroplets on nanostructures.**
Advancing contact angle $\theta_a$ measured by ESEM imaging as a function of the droplet radius $r$ for materials A (nanocylinders, red dots), T4 (truncated nanocones, green dots) and E4 (nanocones, blue dots). In all cases, $\theta_a$ increases and saturates with $r$, but angles are systematically higher by about 20° and 30° on T4 and E4 than on A. Angles on A increase from 110° ± 5° to 140° ± 3° as $r$ varies from 0.6 to 6 µm, on T4 from 150° ± 2° to 163° ± 2° as $r$ varies from 2 to 7 µm, and on E4 from 140° ± 7° to 171° ± 3° as $r$ varies from 0.3 to 7 µm. The solid line is the model for nanocones described in the text and in the Methods section (Eqs. 1 and 2). We also report with dotted lines the contact angles obtained for millimetric water drops on A, T4 and E4, $\theta_a = 167° ± 3°$, 160° ± 2° and 164° ± 3°, respectively. Interestingly, these angles are much larger for A and slightly smaller for T4 and E4 than the saturation value at microscale. Error bars represent standard deviation.

the apex angle is $\beta \sim p/h$ (see Supplementary Fig. 5). Balancing the corresponding pressure $\sim\gamma\beta\,z/p^2$ by the Laplace pressure in the drop $\sim\gamma/r$ yields a depth $z \sim \Sigma\,p^2/r$ (see the Methods section), that is, hyperbolic in drop radius. This formula stresses another advantage of cones, namely their resistance to water penetration[16,31] expressed through the sharpness $\Sigma$. The distance $z$ is nanometric and it quantifies the solid/liquid contact and thus determines the contact angle[34–36], calculated using the Cassie equation[11], $\cos\theta_a = -1 + \phi_s(1 + \cos\theta_0)$, where $\theta_0 \approx 120°$ is the Young water contact angle on hydrophobic silicon, and $\phi_s$ the solid fraction in contact with water. The latter quantity is deduced from the surface areas $A_{ls}$ and $A_{la}$ of the liquid/solid and liquid/air contact whose analytic expressions[35] are given as a function of $z$ in the Methods section (see also Supplementary Figs. 5 and 6). Using our model, we demonstrate the case for E4 (solid line, Fig. 3), where we observe quantitative agreement with the data, explaining in particular why deviations only concern ultra-small drops, below 1.5 µm: above this size, water penetration $z$ becomes negligible. We show further that this limit corresponds to the failure of antifogging. Another more trivial case of failure arises from the cone profile, since $\beta$-angles greater than $2\theta_0 - \pi \approx 60°$ prevent drops from sitting atop the cones[34]. Supplementary Fig. 7 confirms that water invades cones with high apex angle ($\beta = 57° ± 2°$, $\Sigma \approx 1$), which fully inhibits antifogging. In contrast, all our samples have $\beta$-angles between 15° and 38°, which prevents the geometrical impregnation and defines the so-called "sharp cones". All observations and models can be finally put together to build a "phase diagram" of antifogging, as shown in the Supplementary discussion and Supplementary Fig. 8.

The results from Figs. 2 and 3 indicate the differences between the nanostructures, and reveal that droplets can remain in the Cassie state solely for nanocones and truncated nanocones (see also Supplementary Figs. 2 and 4), even on the microscale. In contrast, the behaviour on nanopillars is consistent with previous studies, where condensation induces mixed states[12,17,28,29], as shown in particular by Enright et al. who evidenced pinned wetted areas below microdroplets sitting on needles and pillars[17]. The latter effects are specific to condensing microdrops. Millimetre-size drops deposited on hydrophobic pillars are in a regular Cassie state, as evidenced by the larger value of the contact angle (Fig. 3).

**Antifogging abilities of nanotextures.** We now investigate the antifogging efficiency of nanocones and how it depends on geometry. To achieve this, we visualise the breath figures resulting from condensation on our three families of nanocones and on pillars (see Supplementary Figs. 9 and 10). The experiment relies on lowering the temperature of our samples below the dew point, to typically around 4 °C (Fig. 4a). An inverted microscope equipped with a camera is used to observe how atmospheric water condenses. The supersaturation $S$ (ratio between vapour pressure at room temperature and saturated vapour pressure at sample temperature) is here kept constant at a value $S = 1.6 ± 0.2$.

An experiment lasts 30 min and images, with a size of 700 × 700 µm, are recorded every 2 s. We first observe the nucleation of multiple droplets, with an average density of nuclei per unit area of 1200 mm$^{-2}$ on samples E and H; this value rises to 2300 mm$^{-2}$ on samples T and up to 5600 mm$^{-2}$ on pillars—showing that the presence of flat tops favours nucleation, in agreement with simulations by Xu et al.[18]. In the Supplementary discussion and Supplementary Fig. 11, we further discuss the activity and persistence of the nucleation sites. Nuclei grow and eventually coalesce with their neighbours, and we compare successive images to establish whether a coalescence is followed, or not, by a jump (sketched in Fig. 4a in dark blue). This automated treatment allows us to quantify the jumping rate of a given sample, as a result of statistics performed over the few thousand coalescences that take place within 30 min. A coalescence event implies the merging of $n$ droplets, where $n$ is typically 2–5. The number of events decreases with $n$: the proportion of binary coalescences ($n = 2$) is of the order of 70%, while triple, quadruple and quintuple merging respectively concerns 20%, 6 and 3% of the events. A first overview of material performance can be gained through the global rate $N_g$, defined as the proportion of coalescences resulting in droplet jumps, irrespective of the value of $n$. This quantity is plotted as a function of time in Fig. 4b, where each data point is an average made over 1 min, that is, over ca. 100 coalescences. Considering absolute numbers of events (Supplementary Figs. 12 and 13), it is observed that coalescences and jumps strongly correlate despite their fluctuations (Supplementary Fig. 13), justifying our choice of a rate of jumping as a metric of antifogging.

Despite fluctuations due to the huge polydispersity in drop sizes, the antifogging rate is stationary, with an average value (dotted line) that strongly depends on the texture. As expected from Figs. 2 and 3, where drops were found to be quasi-spherical on cones and adhesive on pillars, we first note an extreme contrast between conical (sample H3, blue dots) and cylindrical texture (sample A, pink dots on the abscissa axis), with respective average values of $N_g = 88\%$ and 0.2%, as also captured through the sharp differences in the breath figures (Fig. S9). Data obtained with all samples of H and E confirm the overall conclusions (Supplementary Figs. 14 and 15), as well as observations performed after increasing the duration of the experiment by a factor of five (Supplementary Fig. 16) or modifying the value of the supersaturation $S$ (Supplementary Fig. 17).

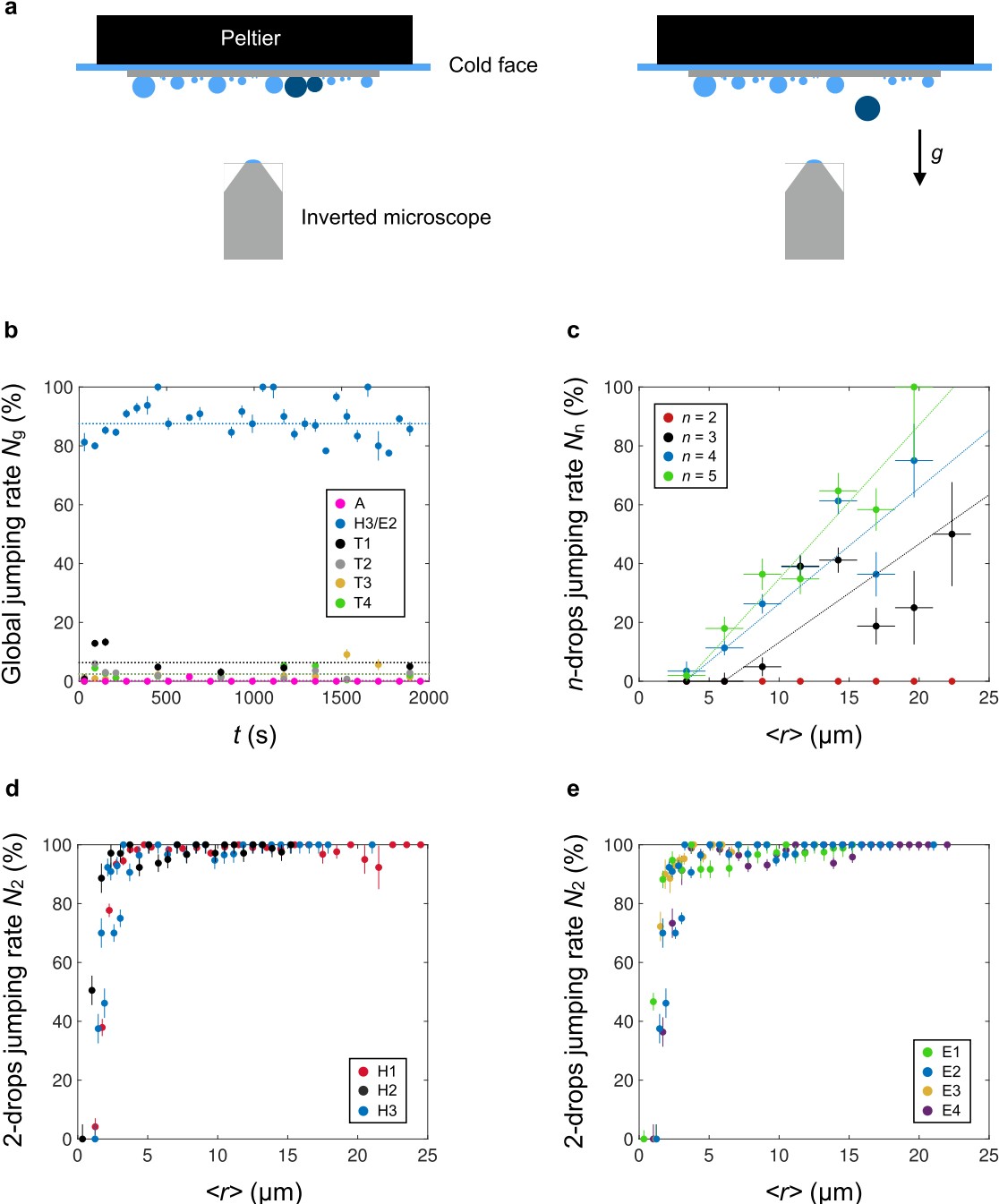

**Fig. 4 Antifogging ability of nanocones. a** Schematic of the experiment: a sample is placed upside down on a Peltier cooler, which generates dew from atmospheric water. Droplets nucleate, grow and eventually coalesce, which we observe with an inverted optical microscope. We focus in particular on coalescing droplets (dark blue) and their possible takeoff from the material, from which we deduce the jumping rate of merging drops. **b** Global jumping rate $N_g$ as a function of time $t$: we consider all observed coalescences and average the proportion $N_g$ that results in droplet jumps after merging, over 1 min. For each series of data (obtained with pillar texture A, conical texture H3/E2 and truncated texture T1 to T4), we indicate with a dotted line the average value of $N_g$. The pink dots on the bottom show the jumping rate $N_g = 0.2\%$ for sample A. **c** Jumping rate $N_n$ of droplets on truncated cones T1 as a function of the mean radius $<r>$ of merging drops, after distinguishing the coalescences that imply $n = 2$, 3, 4, or 5 droplets. $N_n$ increases with both $n$ and $<r>$, explaining why a modest value of $N_g$ can be accompanied by good antifogging abilities. Dotted lines are guides for the eyes. **d**, **e**. Jumping rate $N_2$ on sharp cones H/E for symmetric binary coalescences, for which merging radii do not differ by more than 20%. $N_2$ is plotted as a function of the average radius $<r>$ for homothetic nanocones H1–H3 in **d** and for extruded cones E1–E4 in **e**. In both cases, the jumping rate plateaus at a constant value of 99 ± 1% above a critical radius $r_c \approx 1.5 \pm 0.4\,\mu m$. Error bars represent the standard deviation of data.

Truncated cones provide an intermediate behaviour: $N_g$ decays rapidly with the level of truncation, with an average value of 7% for T1 and of ca. 2% for the samples T2, T3, and T4. Hence truncated cones still have some capacity to repel dew, unlike sample A, despite similar contact-angle hysteresis. We attribute this effect to the conical profile that might still promote nuclei to leave the interspace between structures and sit atop the texture. This interpretation is strengthened by direct ESEM pictures seen in Supplementary Fig. 4: microdroplets on truncated cones are still in a highly hydrophobic state, with a contact angle of ca.160°, a value about 20° higher than on pillars (Fig. 2a).

However, judging the antifogging efficiency of truncated cones solely on the value of $N_g$ can be grossly misleading. While the global performance remains modest compared to that on H and E, breath figures on truncated cones, and especially T1, reveal no accumulation of water after 30 min (Supplementary Movie 1 and Supplementary Fig. 10). Both the fraction occupied by water (~35%) and the radius of the largest drops (~30 μm) are comparable to that observed on nanocones H/E, in sharp contrast with pillars (Supplementary Fig. 9). At first glance, the conjunction of low $N_g$ and efficient water evacuation looks paradoxical. However, we can reconciliate this apparent contradiction by separating the jumping rates $N_n$ for coalescences that imply $n$ drops. These quantities are plotted in Fig. 4c for the sample T1 (and Supplementary Fig. 18 for T4) as a function of the mean radius $<r> = \Sigma\, r_i/n$ of the merging drops, denoting $r_i$ as the sizes of individual drops, with $1 \le i \le n$.

Each data corresponds to typically 40 events for which merging radii differ by no more than 30% for $n > 2$ (see data in Supplementary Tables 2 and 3 for T1 and T4, respectively). Fig. 4c reveals an original antifogging mechanism, compared to the case for sharp cones (blue data in Fig. 4b), where the high $N_g$ implies that droplets jump irrespective of the value of $n$. The jumping rate $N_2$ for binary coalescences ($n = 2$) represents the majority of events, and it is found to be zero on truncated cones— thus explaining the origin for a low $N_g$. However, $N_n$ markedly increases with both $n$ ($n > 2$) and $<r>$, ultimately exceeding 50% for drop radii of ~20 μm at all $n \ge 3$. Smaller droplets suffer more from pining and triple, quadruple and quintuple merging events inject more surface energy than binary merging, which makes it possible to overcome the depinning barrier existing on truncated nanocones. Hence these structures can eventually exhibit a good antifogging ability, yet through a different mechanism than sharp cones: droplets grow for a longer time and are only evacuated when large enough and concentrated enough (which enables multiple coalescences), explaining why the samples are not saturated with water at long time.

We now contrast these results with those of the sharp nanocones H and E for which we focus on the jumping rate $N_2$ (Fig. 4a and Supplementary Fig. 14b), since binary coalescences then inject enough energy to overcome the low water adhesion and generate jumps. Furthermore, we restrict to symmetric coalescences (70% of the binary events), where the ratio between the radii of the two merging drops is between 0.8 and 1.2. As seen in Fig. 4d, e, the corresponding rate $N_2$ is about 100% for $r > 2$ μm, suggesting that the failure of jumping on sharp cones mainly arises from asymmetric merging that fails at injecting enough energy to prompt jumps. We split the results in two graphs that respectively display the antifogging efficiency $N_2$ for homothetic cones (samples H, Fig. 4d) and for extruded cones (samples E, Fig. 4e), both plotted as a function of $<r>$. Each data point is an average over typically 65 coalescences.

Remarkably, all results collapse on a single curve. In all cases, the jumping rate $N_2$ is typically 99% across a broad range of radii (from ~2.5 to ~25 μm, see also Supplementary Figs. 19 and 20),

with a few exceptions at large radius, a case where we have fewer coalescences (typically 10 to 20) so that one sticking event significantly affects the statistics. The very high rate of departure further confirms our assumption that microdroplets remain in the mobile Cassie state, and it generalises the exceptional antifogging character of nanocones reported by Mouterde et al. on a unique sample:[14] the effect is found to be universal across a wide variety of cone geometries.

A second metric for antifogging is the drop radius $r_c$ above which a water drop jumps. This quantity is found to be critical (within only 2 μm in radius $<r>$, $N_2$ varies from 0 to its maximum) and quasi-universal in the explored range of cone geometries. Defining $r_c$ as the size at which we have $N_2 = 50\%$, we find $r_c = 1.8 \pm 0.2$ μm, $1.0 \pm 0.3$ μm and $1.6 \pm 0.3$ μm for samples H1, H2 and H3. For extruded cones E1 to E4, the critical radii are $r_c = 1.1 \pm 0.2$ μm, $1.6 \pm 0.3$ μm, $1.3 \pm 0.3$ μm and $1.9 \pm 0.3$ μm, respectively. These values are fairly constant, with changes comparable to the uncertainty of the measurement—a result also found on cones with similar size, yet convex instead of straight (Supplementary Fig. 21). The typical critical size of jumping nicely agrees with the results in Fig. 3, where the contact angle was found to rapidly decay when the drop radius is below 1.5 μm —an effect we interpreted as resulting from the sinking of water inside the texture (Supplementary Equation 1 and Supplementary Fig. 6). The partial penetration of water in the substrate naturally increases its adhesion and thus impedes the mobility of droplets, preventing them from jumping. Interestingly, as seen in Fig. S20, the quantity $r_c$ was found to be larger on materials with a smaller jumping rate (nanoneedles), confirming the relevance of this parameter for quantifying antifogging.

## Discussion

In summary, the antifogging efficiency of sharp nanocones is found to be universal across a vast range of texture sizes (50–420 nm), apex angles (15–38°) and cone shapes (straight/convex, with sharp/round tips). Drops are observed to be quasi-spherical at microscales, which enables them to jump with a remarkable efficiency. The critical radius $r_c \sim 1.5$ μm of jump corresponds to the drop size at which we record a decrease of superhydrophobicity due to the partial penetration of water in the texture. Upon truncation, cones appear to lose some of their properties, with smaller contact angles and global jumping rates. Nonetheless, condensing water is efficiently evacuated, which brings to light a new antifogging mechanism where, unlike binary merging, jumping is successful for triple, quadruple and quintuple coalescences. The antifogging efficiency also increases with drop size, so that most water can be swept from the surface. This finding should have technological implications: First, it can be desirable to fabricate truncated structures to benefit from their higher mechanical resistance[35], particularly for the case where we predominantly aim at evacuating decamicrometric drops as opposed to smaller ones. Second, sharp cones are likely to wear off over time, and it could previously be anticipated as an irreversible decay of the anti-dew behaviour. However, the clustered departures of drops might favour the persistence of the anti-dew property for blunt or broken tips, at least in the limit where hydrophilic tops (generated by the breaking of hydrophobic cones) play a marginal role. To further advance the understanding, future research might focus on rigorously studying the jumping mechanism: we assumed here that it is related to the penetration of water inside the texture, but the exact threshold remains to be understood, in particular by accounting for the role of contact line pinning at the pillar tops. Another topic of interest concerns the effect of the cone design and chemistry upon the nucleation itself, a mechanism known to be influenced by surface properties, both chemical and physical. A final stimulating question concerns

the increase in texture size: for cones in the micrometric or decamicrometric range, the dew drops will have sizes comparable to that of the texture, which should lead to new regimes of condensation and takeoff, preventing or delaying the antifogging effect.

## Methods

**Surfaces H2, H3, E0, E1, E2, E2', E3, E4, T1, T2, T3, T4.** These materials were produced at University College London according to the following fabrication steps:

(1) A layer of $SiO_2$ (44–100 nm) is deposited on a silicon wafer by plasma-enhanced chemical vapour deposition. The block-copolymer (BCP) poly(styrene-block-2-vinyl pyridine) (PS-b-P2VP) is self-assembled in m-xylene (0.4%) and subsequently spin-casted at 6000 rpm for 30 s resulting in a thin film. The obtained film comprises a well-ordered monolayer of hexagonally packed micelles, in which the molecular weight of each block dictates the distance between neighbouring micelles (pitch).

(2) A polymer breakthrough etch is performed in a PlasmaPro NGP80 Reactive Ion Etcher (RIE) at 20 °C under oxygen plasma in order to remove the polymer matrix. The remaining micellar bumps act as a topographic contrast for the subsequent $SiO_2$ etch.

(3) The micelle pattern is registered into the $SiO_2$ layer using $CHF_3$/Ar plasma etching: RF power 200 W, pressure 50 mTorr, $CHF_3$/Ar 0.3. The $SiO_2$ pattern acts as a hard mask for etching into the underlying Si.

(4) Dry Si etching is performed in an Advanced Silicon Etcher using chlorine plasma under low plasma power in order to achieve slow lateral etching and undercutting of the $SiO_2$ mask. The following conditions are used: Coil power 150–500 W, Platen power 10–60 W, pressure 3–6 mTorr, $Cl_2$ 15–20 sccm.

(5) The remaining $SiO_2$ mask is stripped using hydrofluoric (HF) acid, to produce sharp tipped (H2–H3, E1–E4) or truncated cones (T1–T4), depending on the point at which the etching is stopped.

**Surface A and H1.** In addition, we used as reference samples two materials produced at Brookhaven by A. Checco, A. Rahman and C.T. Black. The surface A is fabricated by combining block-copolymer self-assembly with anisotropic plasma etching in silicon, which provides large-area ($cm^2$) textures with ~10 nm feature size and long-range order. Posts, with diameter $l = 30$ nm and height $h = 88$ nm, are disposed on a rhombus network with side $p = 52$ nm. The surface H1 is fabricated using the same method as for sample A, but etching is made more isotropic, which provides the conical shapes.

**ESEM procedures.** The dynamics of water condensation is imaged using a FEI Quanta 650 field emission gun (FEG) environmental scanning electron microscope of the Laboratoire de Mécanique des Solides at École polytechnique. The sample is mounted on a horizontal bracket for top images and a 60°-tilted copper bracket for tilted images. The support can be inclined up to 90° to provide a clear view of water droplets. The bracket is mounted on a thermoelectric (Peltier) cooling stage and both temperature and chamber pressure are controlled. Before every experiment, five purging cycles are performed, consisting in varying the pressure between 150 and 600 Pa, in order to remove any non-condensable gas. After this procedure, the sample is chilled at around −2 ± 1 °C for 2 min at a vapour pressure of 200 Pa. Water condensation is later achieved by increasing the chamber pressure to about 500–700 Pa. Low beam energies (10 keV) and 3.5 spot size were used to prevent all damage caused by ESEM. A SE detector (GSED) is selected for imaging as it yields better results than BSE detector. Tilting the sample influences the amount of secondary electrons produced, since a greater proportion of the interaction volume is then exposed[37]. Consequently, emission at edges is particularly high and they appear brighter than flat surfaces. The detector potential is set at 330 ± 30 V (bias between 55 and 65) in order to prevent e-beam charging: the electric field magnitude increases with the bias, hence surface potential is more important for high bias. This parameter was found to be crucial for limiting wettability changes during condensation. Higher bias led to the complete wetting of condensing droplets, which might be due to the destruction of the hydrophobic layer. Finally, the electron beam working distance is set around 5 mm. Recordings were performed at various frame rates, varying from 0.3 to 4.6 fps.

**Contact-angle measurements.** Contact angles are deduced from imaging by extracting from image analysis the drop radius $r$ and the contact radius ~ (radius of the apparent contact area of the drop with the surface). The contact angle $\theta$ is simply deduced from the geometric relationship $\sin\theta = \lambda/r$. Since drops arising from condensation are inflating, these experiments provide the so-called advancing contact angle. When slowly inflating a drop, this angle corresponds to the value observed at the contact line when this line starts moving (that is, once the drop is not pinned any more). We denote the velocity of the contact line as $v$. The typical rate of inflation is chosen so as to reach a quasi-static limit for the contact angle, corresponding to capillary numbers $\eta v/\gamma$ (denoting $\eta$ and $\gamma$ as the viscosity and surface tension of water) smaller than $10^{-3}$. In our experiment, we are indeed in the inflating mode, since drops are growing owing to the condensation from the

atmospheric water. The velocity $v$ of the contact line as drops grow (in Fig. 2c, for instance) is between 0.3 and 1 µm/s, so that the capillary number for water is $10^{-8}$, indeed in quasi-static limit.

**Antifogging efficiency of nanotexture: experimental set-up.** The experimental setup is defined in Fig. 4a. The breath figure is observed with a microscope (Nikon Eclipse Ti-U) equipped with a video-camera (Hamamatsu C11440). Samples are placed upside down, so that departing drops do not re-deposit on the material, which would complicate the analysis of the antifogging effect. We can wonder whether gravity might detach droplets (with radius $r$). To that end, we compare its magnitude to the force induced by adhesion by introducing the Bond number Bo ≈ $\rho g r^2/\gamma \sin\theta_a (\cos\theta_r - \cos\theta_a)$. The maximum observed radius of drops condensing on nanocones is 35 µm (owing to the high antifogging efficiency), which yields a Bond number Bo ≈ 0.01 for $\theta_a ≈ 165°$ and $\theta_r ≈ 150°$. Hence gravity can be neglected in our setup, in agreement with the observation that drops never depart without coalescing with their neighbours.

**Arrangement of droplets on nanocones.** Supplementary Fig. 5a sketches the shape of the bottom interface of a droplet in a Cassie state on an array of hydrophobic cones. Contrasting with pillar edges where contact lines can exhibit various contact angles, the contact line on a cone has only one eligible position. The depth $z$ to which water penetrates the texture (Supplementary Fig. 5b) is dictated by the equilibrium between the Laplace pressure inside the droplet and the tension exerted on the contact line. The contact line perimeter being $3\pi b(z)/3$ (Supplementary Fig. 5c), surface tension exerts a force equal to $-\pi b(z)\gamma\cos(\theta_0-\beta/2)$, where we denote $b(z)$ as the contact radius, $\gamma$ as the surface tension of water and $\theta_0 ≈ 120°$ as the Young contact angle of water. Dividing this force by the surface area $A = \sqrt{(3)}p^2/4 - \pi b^2/2$ of the air-water interface (Supplementary Fig. 5c), we deduce a pressure $\Delta P(z)$ that, at equilibrium, balances the Laplace pressure $\Delta P_L = 2\gamma/r$ in the drop. This balance yields a relation between $b(z)$ and $r$, from which we get geometrically the depth $z = 2hb(z)/p$:

$$z(r) = \frac{hr|\cos(\theta_0\beta/2)|}{p}\left[\sqrt{1+\frac{2\sqrt{3}p^2}{\pi r^2\cos^2(\theta_0\beta/2)}}1\right] \quad (1)$$

Hence the depth $z$ is roughly expected to decrease hyperbolically with the radius $r$. For $r \gg p$, Eq. (1) simplifies in $z(r) ≈ \sqrt{(3)}hp/\pi r|\cos(\theta_0 - \beta/2)|$, in agreement with the scaling proposed in the paper. This function is drawn in Supplementary Fig. 6a for the parameters of material E4. Interestingly, the distance $z$ is observed to become significant (approximated as at least 10% of the drop radius) when the drop size is below ~1 µm, in agreement with the observations in Fig. 3.

Using Eq. (1), one can predict the advancing angle $\theta_a$ of a drop with radius $r$ embedded at a depth $z$ in the texture. To that end, we use the Cassie-Baxter model:

$$\cos\theta_a = -1 + \varphi_s(1 + \cos\theta_0) \quad (2)$$

where $\phi_s = A_{1s}/(A_{1s} + A_{1a})$ is the solid fraction in contact with water, a quantity deduced from the areas $A_{1s}$ and $A_{1a}$ of the liquid/solid and liquid/air contact. If we neglect the liquid curvature, we have $A_{1s} = \pi z^2(r) \tan(\beta/2)/\cos(\beta/2)$ and $A_{1a} = [2\sqrt{(3)}h^2 - \pi z^2(r)] \tan^2(\beta/2)$ for a drop with radius $r$ sinking at $z(r)$ and considering a hexagonal array. Using Eq. (1) and the parameters of sample E4, we deduce from Eq. (2) the contact angle $\theta_a$, which we draw with a solid line in Fig. 3. While a qualitative agreement was expected (a smaller drop penetrates further in the texture, so that the increase of solid/liquid contact generates a smaller apparent angle $\theta_a$), the model provides a very satisfactory description of the data—which strengthens our model for drop penetration.

## Data availability

The data that support the plots within this paper and other findings of this study are available in the main text and in the Supplementary Information. Additional information is available from the authors upon reasonable request.

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

## Acknowledgements

We thank Timothée Mouterde and Gaëlle Lehoucq for enlightening discussions, Antonio Checco, Atikur Rahman and Charles Black for providing samples A and H1, Avin Babataheri and Cillian Jezequel for help in the experiments, and Thales for cofunding this project. P.L. thanks the École polytechnique for financial support (Monge Fellowship). S.L. and I.P. thank UKRI/EPSRC for a DTP award grant no EP/N509577/1. I.P., S.L., M.M. and T.L. would like to acknowledge funding from the European Research Council, ERC-StG-IntelGlazing, grant no: 679891 and funding from Lloyd's Register Foundation International Consortium of Nanotechnology (ICON) research grant.

## Author contributions

P.L., I.P. and D.Q. designed the project, S.L., M.M., T.L. and I.P. fabricated samples and discussed the project, P.L. and A.T. performed experiments and analyses, P.L. and D.Q. built the model, P.L. and D.Q. wrote the manuscript with inputs from all other authors.

## Competing interests

The authors declare no competing interests.
