## [Peer Review File · Nature Communications]

REVIEWER COMMENTS

Reviewer #1 (Remarks to the Author):

Review of 'Unique and robust dew-repellency of nanocones' by Quere et al. This work depicts a comparative analysis of condensed droplets across a range of nanostructures which included cylindrical nanopillars, sharp nanocones with varying aspect ratio, as well as truncated nanocone structures. This is an elaborative study related to the corresponding author's 2017 Nature Materials article (Nat. Mater. 2017, 16, 658) where they reported the importance of the nanocone structure for successful anti-fogging. In this manuscript, the authors claim that the sharp nanocones enable the condensed microdroplets to evade the surface by coalescing with a jumping efficiency of nearly 99%. While the truncated nanostructures had a mediocre performance, but also showed successful droplet jumping after multiple coalescences, unlike binary coalescence as in sharp nanocones.

The merits of this work lie in two points:

- (1) Elucidated the design guidelines for anti-fogging surfaces. The antifogging efficiency of sharp nanocones is shown to be universal across a vast range of texture sizes (50 – 420 nm), apex angles (15 – 38°) and cone shapes (straight/convex, with a sharp/round tip). Drops are observed to be quasi-spherical at microscales, which enables them to jump with a remarkable ~99% efficiency.
- (2) Showed broad impacts of jumping droplets on truncated cones. Sharp cones are likely to wear off over time. However, the clustered departures of drops guarantee the persistence of the anti-dew property for blunt or broken tips.

Major comments:

1. Considering the title of the manuscript, how is the robustness of the nanocone structure defined? For what duration of time, the anti-fogging characteristic could be maintained with droplet jumping rate efficiency above 90%?
2. How does the anti-fogging efficiency vary between the homothetic shape, H and the extruded shape, E? Is there a better qualitative method than the 'breath figures under optical microscopy' to differentiate between H and E?
3. In Figure 3, how can the advancing angle be obtained? Advancing angle is defined at the moment when the droplet start moving on a tilted surface. In Figure 3, the droplets did not move. If gravity can be ignored at such a small scale, is that static contact angle?
4. Below 1.5 μm , water penetration z cannot be negligible. Ideally, how can we prevent the penetration of drops with sizes below 1.5 μm ?
5. What will happen if the apex angle β is nearly zero?
6. In Figure 4b for truncated cones, it is shown that for droplet radius slightly higher than 10 μm , the jumping rate for 5 coalescing droplets was lower than either of 3 and 4 coalescing droplets of the same size. This is not in agreement with the usual trend. How would the authors explain that?
7. The truncated cones in this work have hydrophobic coating on the top, but broken cones (the shape is the same as truncated cones) do not have hydrophobic coating on the top. The authors can study truncated cones with hydrophilic top. The fabrication may not be that easy.
8. It'd be interesting to see how the global jumping rate would get affected when the overall size of the homothetic/extruded nanocones be increased further (height greater than 420 nm) to match the size of the nucleating water droplets. If it is hard to do this in this work, it is fine.
9. The effect of the nanocone design on the nucleation of the condensing droplets should also be included for a complete analysis on this area.
10. In this statement "We also note that these effects are specific to condensing drops – larger scale drops sit unambiguously at the top of the structure", can the authors provide direct observation or evidence in this paper?

Some minor comments:

11. The data for truncated nanocones could have been included in the plot for Figure 3 if this type of structure is important in this work.
12. It is clearer to show a schematic of the condensation setup for Figure 4, and put the schematic in the supporting figures.
13. In Figure 4, the Y-axis is clearer to write as "Global jumping rate N_g (%)", not simply " N_g (%)". Others can be changed as well.
14. This sentence is unclear "gravity is an artefact underestimating high contact angles, suggesting that intrinsic angles are rather the ones observed with condensing drops." If it is not the one observed, what is that? Is that larger or smaller? More information is needed for elaboration.

Xianming Dai

Reviewer #2 (Remarks to the Author):

This work integrates couple of components: materials preparation, ESEM measurement, data analysis, for the purpose of antifogging surfaces. Used high impact fabrication techniques, block-copolymer, which has demonstrated the high structural regularity, enabling to link the geometry impact and droplet jumping. This paper utilizes the high-fidelity environmental SEM to measure droplets down to 1.5 μm sizes, which might require an intensive work for analyzing the data.

1. Although the authors introduced a set of surface designs, it is not very clear about the morphological impact on jumping droplet efficiency. The reviewer expects this understanding can be much improved if authors can characterize the shape of nanocones and pillars by introducing universal parameters (or non-dimensional parameter) that represent the pillar or cone morphology in addition to the sharpness of the pillar or nanocones. This work can be more impactful if the paper introduces universal parameters with the demonstration of a complete regime map.

1.1. To the reviewer, the sharpness seems the key to make droplets jump. Is there any way to identify the sharpness by using universal parameters?

1.2. This work can be more useful if the geometries cover a large range of possible morphological parameters, which can create a regime map. The reviewer wonders whether the authors can introduce a universal parameter, create a regime map, to completely cover all possible combinations of p , h , l , sharpness, etc.

1.3. Providing both universal parameters and regime map will help other researchers can use this paper as a design guideline. For example, I make kind of similar pillar structures. With the SEM taken to characterize the geometry of pillars, can I make an appropriate estimation about the droplet jumping phenomena?

2. This paper can be much improved if the phenomena observed here are connected with the jumping mechanisms. Need more quantitative and qualitative explanations about jumping phenomena related to jumping mechanisms.

2.1. For examples, does the jumping recur at the repetitive sites?

2.2. Jumping is a transient, time-dependent process. Are there any impact of increasing time on

jumping phenomena? The current plot showing N_g vs t is not enough, and it does not provide a qualitative explanations.

2.3. The plots in figure 4 show very similar trends for different designs, showing droplet jumpings in %. If this is the case, what is the impact of time (a) geometries (b-d) on the number of jumpings (not %)? Need more explanations to explain the difference of various surface designs. For example, as the time goes, although N_g is 90%, does the number of jumping droplets decrease?

3. The argument about advancing contact angles should be much improved. Could the authors explain the advancing angle mechanisms? Can the authors discuss further about advancing contact angle of other samples? To explain different advancing angles, the authors mentioned pinning / unpinning or full Cassie or partial Cassie. If this is the case, can authors identify those parameters / states by using ESEM or other techniques? What is the criteria?

4. Authors should explain the limitation or more information of ESEM measurement technique. For example, how fast ESEM can capture the images? What is the impact of angled measurement (based on the angle of ESEM, the observation cannot be done from perfect cross-sectional view).

Reviewer #3 (Remarks to the Author):

The quality of jumping-droplet condensation is characterized for a wide variety of nanocone geometries. They find a universal trend of high-quality jumping for superhydrophobic nanocones, regardless of their exact shape, compared to lower-quality jumping for the control case of nanopillars. This high quality is clearly described in two different ways. First, the global percentage of coalescing droplets that are able to jump upon coalescence, which is nearly 90% for nanocones compared to roughly 0% for the nanopillars. Second, the critical size for jumping to occur upon coalescence, which was only 1-2 microns for the nanocone families and rationalized by a pressure balance model. I think the paper is elegant and very well written. However, at the same time, I do have some major concerns particularly with regards to novelty and the nuances of their Cassie model.

Major Comments:

1) Their previous paper in Nature Materials already showed the very high jumping quality of their nanocones. This previous paper even reported the same two measures of jumping quality, i.e. the jumping efficiency and critical size. The primary novelty here is the systematic variation of the nanocone geometry; and yet, they found that these variations did not matter appreciably (aside from shift in critical jumping size for extreme case of truncated nanocones). I do understand that, in addition to the geometric variations here, they expanded more upon the contact angle evolution and conical wetting model, but these things have also been examined to various extents in previous papers on jumping-droplet condensation. Therefore, I am a little concerned that the findings are not quite novel and impactful enough to merit publication in Nature Communications, although I certainly think the paper is well-written and a very nice contribution to the topic overall.

2) I think their model for predicting the solid fraction of the Cassie state for micro-condensate on the nanocones is not comprehensive enough. While their Laplace pressure wetting model is very nice, it nonetheless assumes that the initial location of the droplet's contact line is near the top of the nanocones. However, this ignores the well-reported fact that the majority of embryos would tend to

nucleate from within the nanocone structure, not on the upper tips. The authors therefore need to expand their model by providing some support for how the contact line got up to the top to do their pressure balance in the first place. One possible argument is the preferential nucleation on the tips of the cones, perhaps due to their elevated location in the diffusive boundary layer and/or their sharp tips enhancing diffusion. A second possible argument is that, while the droplets do tend to nucleate from within the structure, they can spontaneously dewet to the top due to a gradient in Laplace pressure and/or wettability along the tapered cones. Yet a third possible argument is that each droplet actually does remain impaled within a single unit cell of roughness, and the Cassie state is due to a subsequent inflation over the neighboring unit cells that would eventually define the outer contact line and contact angle. I do not presume to know which of these three approaches would best answer how the contact line can get to the top in the first place (if it does at all for the nucleating unit cell), but the authors need to resolve this issue for their model to be fully valid.

3) I do not think they are giving the control case of nanopillars a fair assessment. In particular, it is well known that the critical radius for condensate to inflate into a Cassie-like state tends to be at least 10-20 microns for such surfaces. And yet, they only seem to quantify the contact angle for radii less than 10 microns, where inflation has not had a chance to complete. In other words, in Figure 3, what would happen to the contact angle if the x-axis was substantively extended? I'd imagine the angle would continue to increase. I have a similar question for the jumping quality, which was placed at nearly 0% but were the droplets allowed to grow large enough in the experiments? If no jumping was occurring even for larger (~10-100 microns) condensate on the nanopillars, it seems like this is not a fair control case, as many other types of non-tapered nanopillars have been demonstrated to exhibit jumping with an efficiency of at least 30%. Was the critical radius for jumping for the nanopillars even reported here? Was the surface getting flooded?

Minor comments:

4) The authors should give more discussion regarding the importance of the supersaturation on the nucleation density and resulting wetting/jumping conditions. Especially for the nanopillar surface, where I can't help but wonder if the jumping would have been appreciable for smaller supersaturations?

5) I think it would be nice to see more than just two surfaces in Figure 3. While I believe a few more were graphed in the supporting information, it seemed odd to only talk about 2 in the main paper given the focus on systematic variation.

6) When describing the nanocone types on page 2, it was confusing whether the word "variation" was referring to a variation along the heights of a single sample, versus variations between different samples. The pictures helped, but the writing should be more clear.

7) On page 4, and in general, clarify whether the reported hysteresis and/or contact angles are referring to deposited droplets versus dew.

8) Figure 2c, the scale bar is very hard to see.

9) Page 6, the phrase "This strongly suggests a Cassie state triggered at microscales," perhaps clarify at a very small (~1 micron) scale, as even on nanopillared surfaces it is typical to inflate to a Cassie state by 10-20 microns.

10) Page 6 of supporting information, the phrase "the distance z is observed to become significant

when the drop size is micrometric" falsely implies an increasing trend of z with increasing size. At least, it does to me, given that the droplet starts off as nanoscopic.

11) To end on a positive note, I really appreciated their sage comments about how the contact angle of a micro-condensate is higher and more accurate than deposited droplets, due to the minor effect of gravity for the latter case.

REVIEWER COMMENTS

Reviewer #1

Review of 'Unique and robust dew-repellency of nanocones' by Quere et al. This work depicts a comparative analysis of condensed droplets across a range of nanostructures which included cylindrical nanopillars, sharp nanocones with varying aspect ratio, as well as truncated nanocone structures. This is an elaborative study related to the corresponding author's 2017 Nature Materials article (Nat. Mater. 2017, 16, 658) where they reported the importance of the nanocone structure for successful anti-fogging. In this manuscript, the authors claim that the sharp nanocones enable the condensed microdroplets to evade the surface by coalescing with a jumping efficiency of nearly 99%. While the truncated nanostructures had a mediocre performance, but also showed successful droplet jumping after multiple coalescences, unlike binary coalescence as in sharp nanocones.

The merits of this work lie in two points: (1) Elucidated the design guidelines for anti-fogging surfaces. The antifogging efficiency of sharp nanocones is shown to be universal across a vast range of texture sizes (50 – 420 nm), apex angles (15 – 38°) and cone shapes (straight/convex, with a sharp/round tip). Drops are observed to be quasi-spherical at microscales, which enables them to jump with a remarkable ~99% efficiency. (2) Showed broad impacts of jumping droplets on truncated cones. Sharp cones are likely to wear off over time. However, the clustered departures of drops guarantee the persistence of the anti-dew property for blunt or broken tips.

Major comments:

1. Considering the title of the manuscript, how is the robustness of the nanocone structure defined?

We are sorry for the misunderstanding caused by describing the nanocone as “robust”, an ambiguous word since it seems to designate the mechanical robustness of our samples. What we rather had in mind is the universality of the anti-fogging behavior pertaining to a broad range of our samples which vary in the cone geometry and quite surprisingly, includes truncated cones, found to be efficient as well. We therefore modified our title and references in the text to “robustness” and now use instead the word “universal”.

For what duration of time, the anti-fogging characteristic could be maintained with droplet jumping rate efficiency above 90%?

This is indeed an important information to discuss. Following the referee's suggestion, we performed a new series of experiments where we increased the dew formation by a factor of five (about 3 hours instead of 30 minutes). We measured the global jumping rate (percentage of jumps after coalescence) and the area fraction covered by water, as a function of time. After a short transient regime (a few minutes), both curves plateau (stationary regime), showing no degradation of the performance. Interestingly, these new data are superimposable to the old data, showing that the anti-fogging abilities of the sample were maintained for (at least) 18 months, the interval between the two experiments. We added a new figure S16 and comments about this figure in the SI, and also introduced this figure in the main text (in the paragraph following figure 4).

2. How does the anti-fogging efficiency vary between the homothetic shape, H and the extruded shape, E?

We cannot distinguish significant differences between the two families, nor, among each family, when varying the parameters of the design. We measured the global jumping rate as a function of time for all studied samples from families H and E, and calculated the mean value of this rate over time. As displayed and discussed in the SI (section 10, new figure S14a), all values are comparable, which justifies the word “universal” in our title (see item 1). This “asymptote” is understandable when considering the asymmetric coalescences, whose

presence lowers the rate of jumping. When considering symmetric coalescences only, the jumping rate reaches a value close of 100% (new figure S14b), again in a universal way. In addition, the critical radius of jump (a second metric for the anti-fogging efficiency), namely the drop radius for which the jumping rate abruptly goes to zero, is comparable for all samples, within fluctuations due to the relatively small number of events for drop radii smaller than $2\ \mu\text{m}$ (figure S14b).

Is there a better qualitative method than the 'breath figures under optical microscopy' to differentiate between H and E?

Whatever the test we perform, we do not see differences between families E and H (while we see differences with family T). This is true for qualitative tests (breath figure), or quantitative tests (area fraction covered by drops extracted from breath figures (figure S15), or jumping rate and critical radius of jumping, as discussed above.

3. In Figure 3, how can the advancing angle be obtained? Advancing angle is defined at the moment when the droplet starts moving on a tilted surface. In Figure 3, the droplets did not move. If gravity can be ignored at such a small scale, is that static contact angle?

The advancing contact angle can be measured by different means, and the most reliable measurement consists of slowly inflating a drop and looking at the angle in this inflating mode when the contact line (base line) starts moving (that is, once the drop is not pinned any more). The typical rate of inflation is chosen so as to reach a quasi-static limit for the contact angle, corresponding to capillary numbers smaller than 10^{-3} . In our experiment, we are indeed in the inflating mode, since drops are growing owing to condensation from the atmospheric water. The velocity V of the contact line as drops grow (in figure 2c, for instance) is between $0.3\ \mu\text{m/s}$ and $1\ \mu\text{m/s}$, so that the capillary number for water $\eta V/\gamma$ (denoting η and γ as the viscosity and surface tension of water) is 10^{-8} , indeed in quasi-static limit. We modified our text accordingly, see the sentences prior to figure 3.

4. Below $1.5\ \mu\text{m}$, water penetration z cannot be negligible. Ideally, how can we prevent the penetration of drops with sizes below $1.5\ \mu\text{m}$?

From our model (eq. S1), we predict that the water penetration z generally scales as hp/r , denoting h and p as the cone height and pitch, and r as the drop size. In order to minimize the penetration, we need to minimize the product of h by p , that is, to miniaturize even more the texture design. This would imply structures smaller than $100\ \text{nm}$, that is, in a range where van der Waals forces influence the behavior of the liquid, and generally tend to favor the so-called spinodal dewetting of water for hydrophobic materials (an assumption justified by the simulations by Prakash *et al.*, PNAS, 113, 5508 2016). However, it is today a technological challenge to build such a miniaturized texture in a controlled way, at a large scale. With the pitch continuing to decrease, the electrostatic interference from neighboring features becomes more significant, being just one reason to explain the general challenges arising in profile controls, in addition to the stochastic nature of the etchant fluxes. We added this discussion in the SI, section 4 (new paragraph after eq. S1).

5. What will happen if the apex angle β is nearly zero?

We predict that very sharp needles make water penetrate the network of cones (eq. S1 and new figure S6b). Physically, it is due to the fact that a small apex angle cannot prevent water penetration because of insufficient surface area of solid, as it is if β is larger. This explains why angles between 15° and 38° (our range in the study) seem to be optimal for generating anti-fogging – keeping in mind that larger angles also fail due to a complete penetration of water into the texture. We added in the SI, a new paragraph about this case prior to figure S6. It would be interesting to test these ideas by considering cones with a very small apex angle. However, there again, it is a technological challenge to obtain nanocones where $\beta = 1^\circ$, since such cone with diameter of 110 nm (our study) would require a height of 6.3 μm and an aspect ratio of nearly 60!

6. In Figure 4b for truncated cones, it is shown that for droplet radius slightly higher than 10 μm , the jumping rate for 5 coalescing droplets was lower than either of 3 and 4 coalescing droplets of the same size. This is not in agreement with the usual trend. How would the authors explain that?

Thanks a lot for pointing out this inconsistency. We extended the statistical analysis to 35 minutes, which improves the quality of the statistics. This slightly modifies our previous curve and makes the different trends (in particular the jumping rate for 5 coalescing drops) more consistent. However, the number of events remains modest, in particular for “large” drops, which induces significant fluctuations in the data. We now provide a table (new table S2 in section 12 of the SI) that lists all these numbers, which is a useful information for understanding the figure 4c (new name of the former figure 4b). Note finally that the lines drawn in the figure are just guides for the eye.

Using these improved statistics, we now provide a similar plot for the sample T4 (new figure S18b), that is, a much more truncated cone. It is found that the mechanism for droplet ejection we propose is still valid (multiple coalescences are responsible for water evacuation), even if the lower performance of this sample is explained by a shift of the jumping rate towards larger droplet radii. We also provide a table (new table S3) for this new set of data. Furthermore, we added new text in section 12 and introduction sentences in the revised paper for presenting all the new data.

7. The truncated cones in this work have hydrophobic coating on the top, but broken cones (the shape is the same as truncated cones) do not have hydrophobic coating on the top. The authors can study truncated cones with hydrophilic top. The fabrication may not be that easy.

This is a valuable suggestion. The resulting hydrophilic tops should behave as nucleation sites, and thus augment dramatically the number of water nuclei. This favors a quick coalescence, but the pinning of the drops on the truncated cone edges should oppose their depart, and thus delay the anti-fogging behavior. It would be useful to establish the corresponding quantitative laws but this seems difficult today: as mentioned by the reviewer, the fabrication of such samples is not easy since we do not have ways to uniformly break the cones over a large area, nor to modify the surface at a specific location (e.g. solely at the tops of the truncated cones).

8. *It'd be interesting to see how the global jumping rate would get affected when the overall size of the homothetic/extruded nanocones be increased further (height greater than 420 nm) to match the size of the nucleating water droplets. If it is hard to do this in this work, it is fine.*

Thanks again for this suggestion. We plan to do such a study, but the change of scale modifies many major things: 1) The fabrication of either taller structures, or structures with an increased pitch can be achieved. For the former, however, pattern uniformity can begin to suffer. Meanwhile, for the latter, a different masking technique is required (e.g. laser interference lithography as opposed to block-copolymer micelle lithography). 2) The physics of condensation should be different, since we then expect to fill the cell, which might either prevent the depart of the drops or to delay it (only bigger drops might leave, and not small ones as in our study). We conclude our paper with this stimulating suggestion, seen as a natural extension to our work.

9. *The effect of the nanocone design on the nucleation of the condensing droplets should also be included for a complete analysis on this area.*

Thanks for this remark. Following the reviewer's suggestion, we measured the initial density of nucleation on all samples (new section 8 in the SI). Once again, families H and E behave in a similar fashion: it is found that the density of nuclei per unit area is roughly constant on all samples, with an average value of 1200 mm^{-2} (to which corresponds an average distance of about $30 \mu\text{m}$ between nuclei). However, this density changes when tops are present on the texture, that is, on truncated cones and on pillars where the density is, respectively, 2300 mm^{-2} and 5600 mm^{-2} . It was shown in numerical simulations (Xu, Royal Chemistry Society Advances, 5, 812, 2015) that nucleation is prevented at the top of conical structures, which might explain why we observe less nucleation on such samples.

The link between antifogging properties and nucleation map is not obvious: most drops depart with a radius much smaller than $30 \mu\text{m}$, due to the fluctuations in the distance between nucleation sites. However, it is interesting to note that the maximum size of drops departing on cones is roughly $35 \mu\text{m}$, that is, comparable to the mean distance between nucleation sites.

The stationary character of our curves showing the jumping rate as a function of time suggested to look at the nucleation "persistence": we measured the probability of observing a droplet renucleating on a site where a droplet had previously nucleated before taking off. We concentrated on the initial set of nuclei, which defines the "hydrophilic" map of our sample. We consider a population of 205 initial nuclei on sample H1 and measure the number of new nuclei appearing at their location along the condensation experiment. We present in section 8 of the SI (new figure S11) the histogram of renucleation: it is found that the probability of having at least one renucleation is as high as 94%; in addition, these sites are found to be extremely active, since 33% of them will give more than 10 renucleations. We checked that the renucleation probability is comparable on H1 and E4 (with respective values of 94% and 96%). Hence, the initial set of nucleation sites remains "robust" and well-defined throughout the condensation – which contributes to explain the reproducibility of our experiments.

It is worth comparing these numbers with the probability of renucleating on a site occupied by a drop that did not nucleate there. We could observe about 100 droplets on samples H1 and E4 that appeared suddenly at a given "virgin" place of the sample, the result of a coalescence that propels drops along the sample rather than perpendicular to it. We report in the section 8 of the SI respective renucleation probabilities of 7% and 6%, respectively. This low numbers

confirm that nucleation is privileged mainly on active sites, which makes the process repetitive, reproducible and stationary.

We also added sentences in the paragraph prior to figure 4 in the main text where these new observations are introduced.

10. In this statement “We also note that these effects are specific to condensing drops – larger scale drops sit unambiguously at the top of the structure”, can the authors provide direct observation or evidence in this paper?

Thanks for pointing out this confusing sentence. We fully reworded it, and now stress that the different states observed on microdrops and on millidrops are evidenced in figure 3 by the value of the contact angle.

Some minor comments:

11. The data for truncated nanocones could have been included in the plot for Figure 3 if this type of structure is important in this work.

Thanks for the suggestion. It is indeed important to show that the microwetting of truncated cones remain very hydrophobic, a necessary condition for understanding the antifogging abilities of these materials. We modified figure 3 and added data for truncated cones (green data in the figure). We also modified the caption and the text accordingly.

12. It is clearer to show a schematic of the condensation setup for Figure 4, and put the schematic in the supporting figures.

It is indeed very useful to introduce the condensation setup, since it corresponds to a new experiment in the progress of the paper. We now provide the schematic of this setup in figure 4 (new figure 4a) and define in the schematic what we measure, namely the probability of departure after coalescence, which helps to understand the experimental plots in figures 4b, 4c, 4d and 4e. We also wrote a new paragraph (introduction of section 7) in the SI to discuss the role of gravity in the setup.

13. In Figure 4, the Y-axis is clearer to write as “Global jumping rate N_g (%)”, not simply “ N_g (%)”. Others can be changed as well.

Thanks for the suggestion. We adopted the recommended notation.

14. This sentence is unclear “gravity is an artefact underestimating high contact angles, suggesting that intrinsic angles are rather the ones observed with condensing drops.” If it is not the one observed, what is that? Is that larger or smaller? More information is needed for elaboration.

We agree that our formulation was unclear and we fully reworded (and slightly expanded) our explanation of this artefact in the measurement of high contact angles, using millimetric drops.

Reviewer #2

This work integrates couple of components: materials preparation, ESEM measurement, data analysis, for the purpose of antifogging surfaces. Used high impact fabrication techniques, block-copolymer, which has demonstrated the high structural regularity, enabling to link the geometry impact and droplet jumping. This paper utilizes the high-fidelity environmental SEM to measure droplets down to 1.5 μm sizes, which might require an intensive work for analyzing the data.

1. Although the authors introduced a set of surface designs, it is not very clear about the morphological impact on jumping droplet efficiency. The reviewer expects this understanding can be much improved if authors can characterize the shape of nanocones and pillars by introducing universal parameters (or non-dimensional parameter) that represent the pillar or cone morphology in addition to the sharpness of the pillar or nanocones. This work can be more impactful if the paper introduces universal parameters with the demonstration of a complete regime map.

1.1. To the reviewer, the sharpness seems the key to make droplets jump. Is there any way to identify the sharpness by using universal parameters?

We thank the referee for this valuable suggestion. We compare the antifogging abilities of various materials, and it is indeed useful to define dimensionless parameters to classify the observed behaviors. Since the conical structures seem indeed to be key in the antifogging property, the sharpness is a natural quantity to be defined. We define it in the paper as the inverse of the cone angle β ($\Sigma = 1/\beta$, where β is expressed in radian). Σ varies in our study between 1 and 4. The use of Σ helps to classify the different behaviors and show the existence of a domain in sharpness where antifogging can exist: (i) if the sharpness is too low, the water penetrates the cones; (ii) if its is too large, drops should also sink in materials. Hence the antifogging behavior is only observed at intermediate values of Σ .

In the case (ii), the corresponding experiments remain to be done: it is challenging to fabricate regular arrays of cones possessing a sharpness, and therefore aspect ratio, significantly larger than 4, without compromising the uniformity. The increase in aspect ratio at the reduced pitch and feature size (<100 nm) raises some significant problems during etching. Namely, the deeper the etching, the more difficult it is for etchant species to be delivered to the evolving nanocone base as a result of conduction limits and diffusive reflection at the sidewalls (microloading). This reduces the process control and subsequently, we begin to observe inhomogeneities limiting the potential for such a systematic study.

These remarks were added in the revised manuscript and/or in the SI.

1.2. This work can be more useful if the geometries cover a large range of possible morphological parameters, which can create a regime map. The reviewer wonders whether the authors can introduce a universal parameter, create a regime map, to completely cover all possible combinations of p , h , l , sharpness, etc.

1.3. Providing both universal parameters and regime map will help other researchers can use this paper as a design guideline. For example, I make kind of similar pillar structures. With the SEM taken to characterize the geometry of pillars, can I make an appropriate estimation about the droplet jumping phenomena?

This is an excellent suggestion and we answer to these related questions together. The fact that we consider in our study both cones and truncated cones makes potentially the regime map three-dimensional, with independent parameters h , p and l (the texture height, pitch and truncation). Considering that a continuous exploration of this huge “phase diagram” is clearly out of the scope of this study (where we already considered 12 different samples), we chose to build a theoretical phase diagram of the antifogging effect where we include the results obtained with our 12 samples. Another difficulty relies in the presentation of such phase

diagrams, due to the three-dimensionality. With the aim to provide a road map to the designers of antifogging materials, we present two two-dimensional diagrams, that respectively correspond to regular cones and truncated cones. This is the objective of a new section in the SI (section 6), introduced in the main paper at the end of the section presenting the model for water penetration and contact angle. The two-phase diagrams (new figs. S8a and S8) clearly evidence the intervals in the texture parameters where antifogging can be achieved, and help to justify (when we are in this regime) the universal character of our findings. Conversely, the phase diagram helps to predict (for instance from SEM pictures of the samples) whether jumps are likely to occur, or not.

2. This paper can be much improved if the phenomena observed here are connected with the jumping mechanisms. Need more quantitative and qualitative explanations about jumping phenomena related to jumping mechanisms.

2.1. For examples, does the jumping recur at the repetitive sites?

Thanks for this remark. We now provide a complete analysis of the jumping characteristics.

1) We first measure the initial density of nucleation on all samples (section 8 in the SI). Logically, families H and E behave in a similar fashion: the density of nuclei per unit area is roughly constant on all samples, with an average value of 1200 mm^{-2} (to which corresponds an average distance of about $30 \mu\text{m}$ between nuclei). This density changes when tops are present on the texture, that is, on truncated cones and on pillars where the density is, respectively, 2300 mm^{-2} and 5600 mm^{-2} . It was shown in numerical simulations (Xu, Royal Chemistry Society Advances, 5, 812, 2015) that nucleation is prevented at the top of conical structures, which might explain by we observe less nucleation on such samples.

2) As specifically asked by the reviewer, we also discuss, the probability of re-nucleating in a given site of nucleation, after drops departed. We concentrated on the initial set of nuclei, which defines the “hydrophilic” map of our sample. We consider a population of 150 initial nuclei on sample H1 and measure the number of new nuclei appearing at their location along the condensation experiment. We present in section 8 (new figure S11) the histogram of renucleation: it is found that the probability of having at least one renucleation is as high as 96%; in addition, these sites are found to be extremely active, since 35% of them will give more than 10 renucleations. We checked that the renucleation probability is comparable on H1 and E4 (with respective values of 94% and 96%). Hence, the initial set of nucleation sites remains “robust” and well-defined all along the condensation – which contributes to explain the reproducibility of our experiments.

It is particularly worth comparing these numbers with the probability of renucleating on a site occupied by a drop that did not nucleate there. We could observe about 100 droplets on samples H1 and E4 that appeared suddenly at a given “virgin” place of the sample, the result of a coalescence that propels drops along the sample rather than perpendicular to it. As reported in the section 8 of the SI, we find respective renucleation probabilities of 7% and 6%, respectively. This low numbers confirm that nucleation is privileged mainly on active sites, which makes the process repetitive, reproducible and stationary.

All these new results are presented in the new section 8 in the SI. We also added sentences in the paragraph prior to figure 4 in the main text where these new observations are introduced.

2.2. Jumping is a transient, time-dependent process. Are there any impact of increasing time on jumping phenomena? The current plot showing N_g vs t is not enough, and it does not provide a qualitative explanations.

Thanks for asking for this useful clarification. Jumping can be broken down into three phases: (i) Jumping increases, as a result of small nuclei growing and coalescing and subsequently jumping to reach a maximum. (ii) This is followed by a slight decrease in jumping as re-nucleation occurs (new droplets are growing). (iii) Finally, jumping reaches a plateau corresponding to the repetitive sequences of nucleation/growth/departure discussed for instance in question 2.1 above, and in the new section 8 of the SI. Hence we think that the main impact of increasing time on the jumping phenomenon is to set a stationary regime, which explains why we chose to present rates (in %) rather than absolute numbers.

We now discuss this important point in the new section 9 in the SI (with corrections in the main text, as well). As a marker of jumping, we first display (new figure S12) the absolute number of drops on a large sample as a function of time. This number first rises (nucleation phase), then decreases (start of the jumping), and finally reaches a plateau (balance between nucleations and jumps) after typically 10-15 minutes, that is, a short duration compared to the total time of observation (45 minutes). On a sample on which very few jumps are observed (nanopillars), the drop number keeps on decreasing – which corresponds to an increase of the drop sizes.

We also provide in the same section (new figure S13) the absolute numbers of coalescence and jumps for samples H1, H3/E2 and E4 observed for 33 minutes. Data are all similar, and they follow the trend observed in the figure S12 discussed above with a stationary regime at long time. A better way to be convinced by the stationary character of the nucleation/jump process after a transient time can be seen by looking at the area fraction of the sample covered by droplets. It is found to reach a plateau with a value of 35% independent of the sample, providing jumps occur. On nanopillars, this rate can reach values twice larger, due to the progressive invasion of the liquid in the absence of jumps. These different curves are provided in the SI (figure S15).

Following the referee's suggestion, we also performed a new series of experiments where dew formation was increased by a factor of five (about 3 hours instead of 30 minutes). We measured the global jumping rate (percentage of jumps after coalescence) and the area fraction covered by water, as a function of time. After a short transient regime (a few minutes), both curves plateau (stationary regime), showing no degradation of the performance. Interestingly, the new data are superimposable to the old data, showing that the anti-fogging abilities of the sample were maintained for (at least) 18 months, the interval between the two experiments. We added a new figure S16 and comments about this figure in the SI, which we introduced in the main text, in the paragraph following figure 4.

2.3. The plots in figure 4 show very similar trends for different designs, showing droplet jumpings in %. If this is the case, what is the impact of time (a) geometries (b-d) on the number of jumpings (not %)? Need more explanations to explain the difference of various surface designs. For example, as the time goes, although N_g is 90%, does the number of jumping droplets decrease?

Thanks for asking for this essential clarification. As discussed in question 2.2 above, we now provide absolute numbers of drops, coalescences and jumps for three samples decorated by nanocones (new figures S12 and S13). These numbers fluctuate (due to the very nature of the

experiment; for instance, the depart of a big drop can be followed by the nucleation of many little droplets) but the two numbers strongly correlate despite these fluctuations, which justifies the use of the percentage of jumps as a metric of the antifogging efficiency. In addition (to answer a specific question of the reviewer), a plateau for both events is observed at long time, showing that both numbers of events reach a stationary regime.

In addition to the discussion around the new figure S13, we added a sentence in our revised manuscript to emphasize this point (paragraph prior to figure 4).

Interestingly, the number of events is slightly larger on H1, owing to a larger number of nucleations on this sample compared to the other sharp nanocone families. However, if we compare the percentage of jumps N_g (figure S14), they are all similar, which tends to show the universality of the reported phenomena, once we are dealing with sharp nanocones.

3. The argument about advancing contact angles should be much improved. Could the authors explain the advancing angle mechanisms?

We agree that our writing was too quick and even obscure about this important point. The advancing contact angle can be measured by different ways, and the most accurate measurement consists of slowly inflating a drop and looking at the angle in this inflating mode when the contact line starts moving (that is, once the drop is not pinned any more). The typical rate of inflation is chosen so as to reach a quasi-static limit for the contact angle, corresponding to capillary numbers smaller than 10^{-3} . In our experiment, we are indeed in the inflating mode, since drops are growing owing to condensation from the atmospheric water. The velocity V of the contact line as drops grow (in figure 2c, for instance) is between 0.3 $\mu\text{m/s}$ and 1 $\mu\text{m/s}$, so that the capillary number for water $\eta V/\gamma$ (denoting η and γ as the viscosity and surface tension of water) is 10^{-8} , indeed in quasi-static limit. This discussion is summarized in our revised manuscript (see the sentences prior to figure 3).

As also asked by the reviewer, we also provide new data of contact angles. In figure 3, we show the advancing angle of micrometric water drops on truncated nanocones. It is observed that the truncated cones remain very hydrophobic, a necessary condition for understanding the antifogging abilities of these materials. We modified the caption of figure 3 and the text accordingly. These measurements are finally completed by new data of contact angles for two samples from family E (new figure S3) and new data for angles on family T at larger drop radii (figure S4b).

To explain different advancing angles, the authors mentioned pinning / unpinning or full Cassie or partial Cassie. If this is the case, can authors identify those parameters / states by using ESEM or other techniques? What is the criteria?

ESEM cannot visualize water at the nanometric scale between pillars, which makes it impossible to observe directly which state is selected by the condensing droplets. However, it makes it possible to image microdrops (radius larger than 350 nm), and to compare their appearance on the different samples. Figure 2 shows that microdrops on pillars (sample A) and on nanocones (samples E) are markedly different: while angles are on the order of 140° on A, these values jump to 165° to 170° on samples E; this is one of the main results of our study (it is the first time that microdrops are found to have such large angles), and it is an essential step to understand the mobility they have (that is, ability to take off) and the

resulting antifogging abilities. We added a more detailed description of the figure in the revised text.

Hence partial Cassie state (for pillars) and full Cassie state (for cones) are interpretations. These interpretations are reinforced by: 1) our discussion about the values of contact angles of millimetric drops; 2) similar observations in the literature for drops on pillars; 3) the possibility of nanocones to expel the water nuclei from the structures, which provides the basis of our model on contact angles (eqs. S1 and S2) found to fit quantitatively the data without adjustable parameters (figure 3). We partially reworded the text around figure 3 to make this point clearer, and added a whole new paragraph on page 6 about full/partial Cassie states (and relevant references).

The manuscript also includes ESM images of microdrops on truncated nanocones (figure S4a), which show that despite the existence of truncation, angles are significantly higher than on pillars (around 160° instead of 140°), a key observation for understanding that such texture can also be antifogging. This observation is now included in our text.

4. Authors should explain the limitation or more information of ESEM measurement technique. For example, how fast ESEM can capture the images? What is the impact of angled measurement (based on the angle of ESEM, the observation cannot be done from perfect cross-sectional view).

This information was indeed missing. We now added a whole section in the SI (section 2) about the ESEM measurement technique, including specifications about the rate of image capture and about the impact of the tilting angle of ESEM.

Reviewer #3

The quality of jumping-droplet condensation is characterized for a wide variety of nanocone geometries. They find a universal trend of high-quality jumping for superhydrophobic nanocones, regardless of their exact shape, compared to lower-quality jumping for the control case of nanopillars. This high quality is clearly described in two different ways. First, the global percentage of coalescing droplets that are able to jump upon coalescence, which is nearly 90% for nanocones compared to roughly 0% for the nanopillars. Second, the critical size for jumping to occur upon coalescence, which was only 1-2 microns for the nanocone families and rationalized by a pressure balance model. I think the paper is elegant and very well written. However, at the same time, I do have some major concerns particularly with regards to novelty and the nuances of their Cassie model

Major Comments:

1) Their previous paper in Nature Materials already showed the very high jumping quality of their nanocones. This previous paper even reported the same two measures of jumping quality, i.e. the jumping efficiency and critical size. The primary novelty here is the systematic variation of the nanocone geometry; and yet, they found that these variations did not matter appreciably (aside from shift in critical jumping size for extreme case of truncated nanocones). I do understand that, in addition to the geometric variations here, they expanded more upon the contact angle evolution and conical wetting model, but these things have also been examined to various extents in previous papers on jumping-droplet condensation. Therefore, I am a little concerned that the findings are not quite novel and impactful enough to merit publication in Nature Communications, although I certainly think the paper is well-written and a very nice contribution to the topic overall.

We thank the reviewer for his/her balanced report. Our previous paper in *Nature Materials* had indeed showed the great potential of nanoconical structures to repel condensing drops, but the microwetting of water on such materials was not described neither modeled. In addition, experiments were performed on a single sample, which made the message quite fragile. In our current paper, we do not only extend the effect to families of cones, we also consider a new structure (truncated cones) for which we could anticipate a strong reduction of the antifogging ability, and where, however, it is found that microdrops continue to be efficiently swept away – owing to a new scenario of antifogging.

In summary: 1) Our first finding is the universality of the phenomenon, as pointed out by the referee. This was needed to generalize what was known, but also had to be established for practical applications. 2) The second novelty is the measurement of contact angles at the scale of microdrops, and the significant observation of quasi-spherical shapes in a limit where droplets are rather expected to be “sticky”. Our pictures show that the materials are systematically non-wetted by microdrops, which explains the unprecedented jumping rates on these surfaces. 3) The third novelty is the observation and interpretation of high jumping rates with truncated nanocones.

Figure 1 presents our samples and the necessity of probing families of materials (with the production of three controlled families of samples obtained by very new techniques of nanofabrication). Figures 2 and 3 show and discuss the aforementioned original item 2. Specifically, Fig 2b has no equivalent in the literature, and despite looking simple, both the material fabrication and the imaging technique are highly challenging. Fig 4 shows the universality of the antifogging behavior of cones and also the novel antifogging mechanism for truncated cones, and it is thus mostly original. However, we understand the concern in the referee’s report and tried to emphasize more clearly the novelties of our paper, from the title to the new figures passing by the writing, but also by reinforcing significantly the supplementary information (new discussions on the nucleation processes and recurrences on nanocones, microwetting of other samples, establishment of phase diagram of jumping, etc.).

2) I think their model for predicting the solid fraction of the Cassie state for micro-condensate on the nanocones is not comprehensive enough. While their Laplace pressure wetting model is very nice, it nonetheless assumes that the initial location of the droplet's contact line is near the top of the nanocones. However, this ignores the well-reported fact that the majority of embryos would tend to nucleate from within the nanocone structure, not on the upper tips. The authors therefore need to expand their model by providing some support for how the contact line got up to the top to do their pressure balance in the first place. One possible argument is the preferential nucleation on the tips of the cones, perhaps due to their elevated location in the diffusive boundary layer and/or their sharp tips enhancing diffusion. A second possible argument is that, while the droplets do tend to nucleate from within the structure, they can spontaneously dewet to the top due to a gradient in Laplace pressure and/or wettability along the tapered cones. Yet a third possible argument is that each droplet actually does remain impaled within a single unit cell of roughness, and the Cassie state is due to a subsequent inflation over the neighboring unit cells that would eventually define the outer contact line and contact angle. I do not presume to know which of these three approaches would best answer how the contact line can get to the top in the first place (if it does at all for the nucleating unit cell), but the authors need to resolve this issue for their model to be fully valid.

We thank the referee for asking for this useful clarification. Our paper was too impressionistic on this important point, and we now discuss the origin of a Cassie state in a new paragraph on page 6. We consider different possibilities (in the spirit of the reviewer's assumptions) and discuss the scenario of dewetting. We also cite four papers where related discussions were made.

The new paragraph discusses why cones are so special: As shown numerically, nucleation should energetically be favoured on the cone sides and bottoms, and embryos growing there leave air at the base of the cone (a form of spinodal dewetting discussed in simulations and observed experimentally). The droplet curvature remains constant as long as its size remains smaller than cone-to-cone distance (around 100 nm in our experiments), but microdrops (the ones we observe) sterically interact with several cones, forcing them to host unbalanced Laplace pressure which drives them to the top. Hence droplets then do not anchor in the texture (the definition of a Cassie state), in agreement with observations made by looking at the condensation of hot water: no adhesion was measured between water and the conical substrate, independently of the drop temperature, while drops with a foot in the texture would stick to it, and more pronounced as temperature is increased. Hence, we can interpret the unique behaviour of microdrops on nanocones (figure 2b) as resulting from a spontaneous depinning of water nuclei from the core of such a texture.

3) I do not think they are giving the control case of nanopillars a fair assessment. In particular, it is well known that the critical radius for condensate to inflate into a Cassie-like state tends to be at least 10-20 microns for such surfaces. And yet, they only seem to quantify the contact angle for radii less than 10 microns, where inflation has not had a chance to complete. In other words, in Figure 3, what would happen to the contact angle if the x-axis was substantively extended? I'd imagine the angle would continue to increase.

I have a similar question for the jumping quality, which was placed at nearly 0% but were the droplets allowed to grow large enough in the experiments? If no jumping was occurring even for larger (~10-100 microns) condensate on the nanopillars, it seems like this is not a fair control case, as many other types of non-tapered nanopillars have been demonstrated to exhibit jumping with an efficiency of at least 30%. Was the critical radius for jumping for the nanopillars even reported here? Was the surface getting flooded?

Again, we thank the reviewer for asking this series of questions. This made us extend the discussion about nanopillars, as requested (new section 13 in the SI and modifications in the main text). 1) We now provide a curve of measured contact angles as a function of the condensing drop radius r where r is increased by 50% on pillars, and by 100% on cones (these limits correspond to the end of the experiment in both cases) (figure S19a). In both cases, we observe a plateau for the contact angle, which suggests that there is no change in the state adopted by the droplets. In the literature, the plateau is reached when the drop size is typically

10 times the spacing between pillars, which (due to the nanospacing in our pillars) makes us anticipate a plateau for drops of a few micrometers only – and thus makes us understand why we observe a plateau. 2) We also provide a curve of the jumping rate as a function of the drop radius on cones (up to $r = 30 \mu\text{m}$, above which there is no data since drops cannot grow when antifogging is efficient) and on pillars (up to a radius of $65 \mu\text{m}$, in the range of $10\text{-}100 \mu\text{m}$ asked by the reviewer) (figure S19b). In the latter case, drops do not jump whatever the drop size, which confirms the absence of mobility of these droplets, in agreement with their smaller contact angle. At long time, the surface is not flooded since we clearly distinguish large individual drops, but rather occupied by “very large” drops (radius of $150 \mu\text{m}$ after 45 minutes) that show the inability of the sample to self-evacuate water (figure S9, first line).

The considered pillar sample is unusual, having a nanometric size (both in spacing and in height), which might explain differences with other samples reported in the literature. However, it is important to consider this sample to show unambiguously the differences between samples differing only by their shape (samples A and H1).

But the reviewer is also right to emphasize that other “pillar-like” samples might have antifogging properties. To the best of our knowledge, all these samples have a needle-like structure, and the combination of a slender shape and hydrophobicity by virtue of the “dewetting” from the structures (for instance, if the needles are flexible, they can expel water nuclei). We added this case to our discussion, because: 1) it is important to connect what we did with what was done in the literature; 2) these needle-samples were the ones for which the highest antifogging efficiency was found, which makes clearer the advantage of using nanocones, found to be much more efficient (by a factor three for the antifogging rate) and mechanically more robust.

We report in the section 13 of the SI data extracted from the paper in the literature where we found the highest jumping rate reported on needles (about 36% for binary coalescences, reference 17 in our bibliography). We analyzed the movie provided by the authors of this study and the new figure S20 shows whether a drop departs after a binary coalescence, as a function of its size. The sample consists of nanopillars with a height of 600 nm , an interpillar distance of 300 nm and an aspect ratio of 6. Jumping can occur if the drop radius is large enough (say, above $5 \mu\text{m}$), but the number of observed coalescences (less than 100) makes it difficult to establish statistics for each size, as done in the figure 4 where we analyze thousands of coalescences.

The scenario of jumping on the needles remains to be established. The increase of jumping might agree with the observation that the contact angle increases with the drop size – but the absence of a model for jumping as a function of contact angle makes it difficult to go beyond this qualitative observation.

Minor comments:

4) The authors should give more discussion regarding the importance of the supersaturation on the nucleation density and resulting wetting/jumping conditions. Especially for the nanopillar surface, where I can't help but wonder if the jumping would have been appreciably for smaller supersaturations?

Thanks for the suggestion. We did new experiments and report in the new figure S17 (and section 11 of the SI) the influence of supersaturation on the antifogging efficiency for nanocones and nanopillars. We first observe that the nucleation density logically decreases at

lower supersaturation. We also report that the jumping rate scarcely depends on the supersaturation. For nanocones, we surprisingly observe an increase of antifogging when decreasing supersaturation, which might be due to a decrease of asymmetric coalescences, the main source of jumping inhibition. For nanopillars, the antifogging abilities remain quasi-null for three values of supersaturation, which shows the inability of material A to repel dew at any temperature.

5) I think it would be nice to see more than just two surfaces in Figure 3. While I believe a few more were graphed in the supporting information, it seemed odd to only talk about 2 in the main paper given the focus on systematic variation.

Thanks for the suggestion. We now report in the figure 3 all the main families of texture studied in the paper (one sample per family for the sake of clarity), pillars, cones, truncated cones. We also added a new figure S3 with more data on nanocones.

6) When describing the nanocone types on page 2, it was confusing whether the word “variation” was referring to a variation along the heights of a single sample, versus variations between different samples. The pictures helped, but the writing should be more clear.

Sorry for the confusion. We slightly reworded our writing.

7) On page 4, and in general, clarify whether the reported hysteresis and/or contact angles are referring to deposited droplets versus dew.

Thanks for pointing out this source of confusion. We reworded the corresponding sentence and also made clearer why we access advancing angles in the dew experiments.

8) Figure 2c, the scale bar is very hard to see.

Thanks for noting it. It is now more visible.

9) Page 6, the phrase “This strongly suggests a Cassie state triggered at microscales,” perhaps clarify at a very small (~1 micron) scale, as even on nanopillared surfaces it is typical to inflate to a Cassie state by 10-20 microns.

We followed the reviewer suggestion and reworded the corresponding sentence.

10) Page 6 of supporting information, the phrase “the distance z is observed to become significant when the drop size is micrometric” falsely implies an increasing trend of z with increasing size. At least, it does to me, given that the droplet starts off as nanoscopic.

We agree that there is a source of confusion. We corrected it, and also defined what we mean by “significant” (otherwise a bit obscure).

11) To end on a positive note, I really appreciated their sage comments about how the contact angle of a micro-condensate is higher and more accurate than deposited droplets, due to the minor effect of gravity for the latter case.

Thanks for the positive comment. We expanded slightly this remark by incorporating our findings with truncated cones (new figure 3), which reinforce our interpretation.

REVIEWERS' COMMENTS

Reviewer #1 (Remarks to the Author):

The authors have addressed all the comments. I recommend accepting as it is.

Reviewer #3 (Remarks to the Author):

The authors did a tremendous job with this revised manuscript and rebuttal. I especially appreciate the articulate clarifications of the novelty here with respect to their prior Nature Materials paper, the more in-depth analysis of comparing condensation/jumping across the various surfaces, and more clearly explaining why the nanopillars perform so poorly. I now see that their point was that most of the previous jumping-droplet papers did indeed utilize a nanostructure that had some level of tapered shape to the features, which this present paper is elucidating is crucial. Indeed, when they fabricate perfect nano pillars with no taper, the performance is surprisingly poor which was not obvious from previous reports. They also better explained how the droplets managed to shift from the bottom of units cells.

Therefore, if the other reviewers are similarly satisfied, I can now recommend this paper for publication in Nature Communications.